# *IL-10* Gene Polymorphisms and IL-10 Serum Levels in Patients with Multiple Sclerosis in Lithuania

**DOI:** 10.3390/brainsci12060800

**Published:** 2022-06-18

**Authors:** Ugne Masilionyte, Greta Gedvilaite, Kriste Kaikaryte, Alvita Vilkeviciute, Loresa Kriauciuniene, Brigita Glebauskiene, Renata Balnyte, Rasa Liutkeviciene

**Affiliations:** 1Medical Academy, Lithuanian University of Health Sciences, Eiveniu Street 2, LT-50161 Kaunas, Lithuania; 2Neuroscience Institute, Medical Academy, Lithuanian University of Health Sciences, Eiveniu Street 2, LT-50161 Kaunas, Lithuania; greta.gedvilaite@lsmuni.lt (G.G.); kriste.kaikaryte@lsmuni.lt (K.K.); alvita.vilkeviciute@lsmuni.lt (A.V.); loresa.kriauciuniene@lsmuni.lt (L.K.); rasa.liutkeviciene@lsmuni.lt (R.L.); 3Department of Ophthalmology, Medical Academy, Lithuanian University of Health Sciences, Eiveniu Street 2, LT-50161 Kaunas, Lithuania; brigita.glebauskiene@lsmuni.lt; 4Department of Neurology, Medical Academy, Lithuanian University of Health Sciences, Eiveniu Street 2, LT-50161 Kaunas, Lithuania; renata.balnyte@lsmuni.lt

**Keywords:** multiple sclerosis, *IL-10* gene polymorphisms, IL-10 serum levels

## Abstract

Multiple sclerosis (MS) is a chronic inflammatory disease of the central nervous system with features of demyelination and axonal degeneration at a young age. Genetic factors may play an important role in the development of multiple sclerosis. (1) **Objective:** To investigate *IL-10* rs1800871, rs1800872, rs1800896, and IL-10 serum levels in patients with multiple sclerosis. (2) **Methods:** Our study included patients with multiple sclerosis (*n* = 127) and healthy volunteers (*n* = 195). The subjects’ DNA was extracted from peripheral blood leukocytes and genotyped by real-time polymerase chain reaction. The results were analyzed using the program “IBM SPSS Statistics 27.0”. (3) **Results:** The *IL-10* SNPs were analyzed between the MS and control groups; however, no statistically significant results were found. The serum levels of IL-10 in the groups of MS and healthy subjects were not statistically significantly different (median (IQR): 0.828 (1.533) vs. 0.756 (0.528), *p* = 0.872). (4) **Conclusions:** *IL-10* rs1800871, rs1800872, and rs1800896 and serum IL-10 levels are not likely to be associated with MS development. However, individuals carrying the rare haplotypes of rs1800871, rs1800872, and rs1800896 were associated with increased odds of MS (*p* = 0.006).

## 1. Introduction

Multiple sclerosis (MS) is one of the most common diseases of the central nervous system (CNS), mainly affecting young people [1]. The prevalence of this disease is increasing, especially in women, and varies by race and geographic region [2,3]. MS is a leading cause of disability in young adults in Europe and the United States [4]. The main cause of this disease is still unknown, but epigenetic, genetic, and environmental factors contribute to the risk of developing MS [5]. Immune cells (T and B lymphocytes, macrophages, plasmocytes, cytokines, etc.) are involved in the pathogenesis of MS. According to the theory of molecular mimicry, these cells cause inflammation, active damage, and tissue destruction through a reaction triggered by the immune system against its own myelin protein [6]. Over the past decade, researchers have identified a number of genetic factors that increase the susceptibility to MS in the general population [7]. The *IL-10* gene is located on chromosome 1q32.1. The gene consists of five exons that encode the IL-10 proteins of 178 amino acids. The *IL-10* gene is activated by the receptors IL-10-R1 and IL-10-R2. Decreased IL-10 mRNA expression is observed at MS, which may be associated with promoter polymorphisms [8]. Three single-nucleotide polymorphisms, rs1800871, rs1800872, and rs1800896, are associated with alterations in IL-10 cytokine production. IL-10 is an important cytokine regulator of the immune system, secreted mainly by macrophages and T lymphocytes. It is involved in the immune response to infectious and autoimmune diseases. IL-10 has an anti-inflammatory effect by reducing the production of immunoactive molecules (TNF-α, IFNγ, and IL-12) and inhibiting antigen-specific cytotoxic T cells. Since IL-10 may have an inflammatory function, it activates B cells and promotes autoantibody production. In the absence of IL-2, it inhibits T-cell apoptosis and promotes T-cell growth [9]. Low IL-10 production is observed before relapse in chronic remitting MS, and increased IL-10 levels correlate with disease remission [10]. It has been shown to increase the productivity of IL-10-producing cells [11]. Therefore, the aim of our study was to investigate *IL-10* rs1800871, rs1800872, and rs1800896 and IL-10 serum levels in patients with multiple sclerosis in Lithuania.

## 2. Materials and Methods

### 2.1. Study Subjects

The study was approved by the Ethics Committee for Biomedical Research, Lithuanian University of Health Sciences (no. BE-2-/102). The study was conducted in the Department of Ophthalmology and the Department of Neurology, Lithuanian Health Sciences University Hospital (Kaunas, Lithuania). All subjects who participated in the study provided a written informed consent form. During the research, two groups were formed. Group I: patients with multiple sclerosis (*n* = 127). Group II: healthy subjects (*n* = 195). Patients were excluded if they had other systemic illnesses (diabetes mellitus, oncological diseases, systemic tissue disorders, chronic infectious diseases, autoimmune diseases, conditions after organ or tissue transplantation), obscuration of the eye optic system, or because of poor fundus photography quality. Diagnosis of MS was confirmed with 2017 diagnostic criteria: by clinical symptoms/relapse, brain/spinal cord MRI (Magnetic Resonance Imaging) findings with typical demielinating lesions (according to MAGNIMS criteria) and positive oligoclonal bands [12,13].

### 2.2. Polymorphism Selection

The rs1800871, rs1800872, and rs1800896 SNPs are located in the upstream *IL-10*-promoter region and associated with transcription of IL-10 mRNA and IL-10 protein expression in vitro [14]. The eligible studies for meta-analysis were identified by an electronic search of the published literature using databases such as PubMed, Web of Science, Science Direct and UpToDate to obtain related articles published between 1999 and September 2018. The articles were considered based on specific (inclusion and exclusion) criteria. The inclusion criteria for selection of studies were (i) case-control study design and (ii) genotype frequencies available for both cases and controls. The exclusion criteria were (i) studies conducted without control subjects, (ii) studies performed on animal models, (iii) studies from cell lines and case reports, and (iv) studies with insufficient genotypic data.

### 2.3. The DNA Extraction and Genotyping

After collecting the venous blood samples (white blood cells), the DNA salting-out method was used for preparing genomic DNA.

The genotyping of *IL-10* rs1800871, rs1800872, and rs1800896 was carried out using the real-time PCR (Polymerase Chain Reaction). Three SNPs were genotyped on the Step One Plus real-time PCR system (Applied Biosystems, Chicago, IL, USA). The TaqMan^®^ SNP genotyping assays (Thermo Scientific, Waltham, MA USA) for all SNPs were performed according to the manufacturer’s protocol. The Allelic Discrimination program was used during the real-time PCR. The program determined individual genotypes according to the fluorescence intensity rate of different detectors (VIC and FAM). DNA was isolated from peripheral-venous blood leukocytes using salting-out method. The method is based on collecting cells by centrifugation, their suspension in a buffer solution, the degradation of cell membranes with detergents, the hydrolysis of proteins by proteinase K, and the deproteinization chloroform, and the precipitation of DNA with ethanol. Validated assays were used for real-time PCR:rs1800871 Context Sequence [VIC/FAM]:

AGTGAGCAAACTGAGGCACAGAGAT[A/G]TTACATCACCTGTACAAGGGTACAC

rs1800872 Context Sequence [VIC/FAM]:

CTTTCCAGAGACTGGCTTCCTACAG[T/G]ACAGGCGGGGTCACAGGATGTGTTC

rs1800896 Context Sequence [VIC/FAM]:

TCCTCTTACCTATCCCTACTTCCCC[T/C]TCCCAAAGAAGCCTTAGTAGTGTTG

### 2.4. Quantification of IL-10 Serum Levels

IL-10 serum levels were measured in 19 control subjects and 26 patients with MS. The assay was determined by ELISA (enzyme-linked immunosorbent assay) using the Invitrogen IL-10 Human ELISA Kit. IL-10 serum levels standard curve sensibility range: 7.8–500 pg/mL, sensitivity <1 pg/mL. Serum levels were analyzed following the manufacturer’s instructions, using a Multiskan FC Microplate Photometer (Thermo Scientific, Waltham, MA, USA) at 450 nm. The samples were excluded if the levels of serum were below the detection range.

### 2.5. Statistical Analysis

Statistical analysis was performed using the Statistical Package for the Social Sciences, version 27.0 for Windows (SPSS for Windows—Statistical Package for the Social Sciences for Windows, Inc, Chicago, IL, USA, version 27.0). The hypothesis of a normal difference in the values of the measured symptoms was tested using the Kolmogorov–Smirnov and Shapiro–Wilk tests. The characteristics of the subjects did not meet the criteria of the normal distribution. Therefore, the following descriptive statistical characteristics were used: median, interquartile range (IQR), mean rank.

The distribution of genotypes of polymorphisms in the study groups was assessed according to the HWE (Hardy Weinberg Equilibrium) equilibrium model. The χ^2^ test and the Fisher test were used to compare the homogeneity of the polymorphism distribution of *IL-10* rs1800871, rs1800872, and rs1800896. In the binary logistic regression analysis, the odds ratio (ORs) with 95% confidence interval (CI) of MS was estimated, considering inheritance patterns and combinations of genotypes. The selection of the best genetic model was based on the Akaike Information Criterion (AIC). Therefore, the best genetic models were those with the lowest AIC values. To test the statistical hypotheses, we selected a criterion significance level of 0.05. A statistically significant difference was found when the *p* value was <0.05. Mann–Whitney U test was used to compare IL-10 concentrations in different groups. Differences were considered statistically significant when the *p* value was <0.05.

*IL-10* haplotype association analysis was performed on the patients with multiple sclerosis and control groups separately. We used online SNPStats website (https://www.snpstats.net/snpstats/ (accessed on 13 April 2022)). Pairwise linkage disequilibrium (LD) analysis was assessed by D’ and r^2^ measures. The associations between the haplotypes and multiple sclerosis were calculated by logistic regression and presented as ORs and 95% CI. All haplotypes with frequencies less than 1% were merged into one group and described as “rare” haplotypes.

## 3. Results

In total, 322 individuals were included in the study. Two groups of subjects were formed to study the genotype distribution of single-nucleotide polymorphisms in the *IL-10* gene. The first group consisted of 127 people with multiple sclerosis, of whom 44 (34.6%) were men and 83 (65.4%) were women. The median age of this group was 36 years. The control group consisted of 195 individuals, 49 men (25.1%) and 146 women (74.9%). The demographic data of the subjects are presented in Table 1. There were no differences between women and men (*p* = 0.058) and no differences in the age of the study groups (*p* = 0.054).

For the *IL-10* rs1800871, rs1800872, and rs1800896 genotype frequencies in the multiple sclerosis and healthy population groups, a Hardy–Weinberg analysis was performed to compare the observed and expected frequencies of the *IL-10* rs1800871, rs1800872, and rs1800896 using the *χ^2^* test in the control group. The genotype distribution of the polymorphisms matched the Hardy–Weinberg equilibrium (*p* > 0.001). The *IL-10* rs1800871, rs1800872, and rs1800896 genotype and allele frequencies did not significantly differ between the MS and control groups. The results are shown in Table 2.

A binomial logistic regression analysis of the *IL-10* rs1800871, rs1800872, and rs1800896 genotypes in the MS patients and control groups was performed, but we found no statistically significant results (Appendix A).

Furthermore, we compared the *IL-10* rs1800871, rs1800872, and rs1800896 genotype and allele frequencies between the patients with MS and the control-group subjects according to age. Two groups were formed, age ≤30 and age >30, but the prevalence groups did not differ between them (Table 3).

A further statistical analysis was performed to evaluate the *IL-10* rs1800871, rs1800872, and rs1800896 associations with MS development in the men and women separately. However, no differences were found between the men or women with MS and the controls (Table 4). A binary logistic regression analysis was performed for the males and females separately. Unfortunately, no statistically significant results were found (Appendix A).

### 3.1. IL-10 Serum Levels in MS and Control Groups

Serum IL-10 levels were measured in groups of patients with MS (*n* = 26) and healthy subjects (*n* = 19). Comparing the IL-10 serum levels between the two groups, no statistically significant differences were observed (median IQR): 0.828 (1.533) vs. 0.756 (0.528), *p* = 0.872) (Figure 1).

Considering the associations between the SNP genotypes and the IL-10 levels, we analyzed the IL-10 levels in the different genotype groups, but no statistically significant differences were found (Table 5).

### 3.2. IL-10 Haplotype Analysis

A haplotype analysis was performed on the multiple sclerosis and the control groups. A pairwise linkage disequilibrium (LD) between the studied IL-10 SNPs was observed (the results are shown in Table 6).

D’ is the deviation between the expected haplotype frequency and the observed frequency (D’ scale: 0,1), R2 is the squared correlation coefficient of the haplotype frequencies (r^2^ scale: 0,1), and p is the significance level when *p* = 0.05.

The haplotype frequency analysis did not show any associations with MS (Table 7) but revealed that the individuals carrying rare haplotypes of rs1800871, rs1800872, and rs1800896 were associated with increased odds of MS (OR = 5.99, 95% CI: 1.68–21.34; *p* = 0.006) (Table 7).

## 4. Discussion

The aim of our study was to find an association between *IL-10* rs1800871, rs1800872, and rs1800896 and IL-10 serum levels in patients with multiple sclerosis in the Caucasian population. These polymorphisms have been previously investigated in studies examining the pathogenesis mechanism of MS, but the results remain controversial. We found that the *IL-10* rs1800896 T allele might play a predominant role in the development of MS in women. No statistically significant differences were found in the male group. The binary logistic regression performed to evaluate the effect of these polymorphisms on the occurrence of MS in males and females did not yield statistically significant results.

Various findings on these polymorphisms are also found in studies by other authors. Luomala et al. found that the *IL-10* gene rs1800896 AG genotype protects against the severe form of MS (*p* = 0.010), and that the effect increases over time (after 10 years *p* = 0.043, after 15 years *p* = 0.025). This polymorphism affects the severity of the disease rather than its onset [15]. A study performed of 152 MS patients and 242 healthy subjects by Asgharzadeh et al. found that the *IL-10* rs1800896 AA genotype was more common in the control group (*p* = 0.021) and the rs1800896 AG genotype was more common in the patients with MS (*p* = 0.015). It is suggested that the rs1800896 AG genotype, combined with changes in interleukin production and environmental factors, is a risk factor for the development of MS, and the *IL-10* rs1800896 AA genotype may be considered a protective factor for MS [16]. Naseri et al. studied the SNP of the *IL-10* gene (rs1800871, rs1800896, and rs1800872). Only the *IL-10* gene rs1800872 showed a significant difference between the MS patients and the control group (*p* = 0.02) [17]. Mihailova et al. included 55 MS patients and 86 healthy subjects in their study. It was found that the *IL-10* rs1800871 and rs1800872 CC genotypes were statistically more frequent in patients with MS (*p* = 0.015). The uniform distribution of the genotypes of these polymorphisms can be explained by the strong linkage disequilibrium of their association. The significantly increased distribution of the rs1800871 and rs1800872 CC genotypes in patients with MS may be associated with low IL-10 protein expression. This suggests a disturbance in the regulation of T2 helper cells (Th2) during the immune response in MS [18]

In this study, we examined the distribution of the genotypes and alleles of these polymorphisms in the study groups. We found no statistically significant difference between the MS group and the control group. The binary logistic regression analysis of the *IL-10* rs1800871, rs1800872, and rs1800896 SNP did not yield statistically significant results. Therefore, there were no statistically significant differences between the age groups (up to 30 years and 30 years and older).

We agree with the results of many other studies, which analyzed the association of the *IL-10* gene rs1800871, rs1800896, and rs1800872 polymorphisms with MS. Azarpira et al. included 110 patients with MS and 100 healthy subjects in their study. When examining the distribution of the genotypes and alleles in the groups of healthy and MS patients, the researchers found no statistically significant differences: rs1800871, rs1800872, and rs1800896 had values of *p* = 0.85, *p* = 0.85, and *p* = 0.95, respectively [19]. Therefore, de Jong et al. investigated the association of the *IL-10* gene rs1800896, rs1800871, rs1800890, rs6703630, and rs6693899 SNP with clinical manifestations of MS. Statistically significant data were found only with the rs6703630 polymorphism (*p* = 0.04) in the healthy group. The present study revealed that the mean endotoxin-induced IL-10 production in whole-blood samples was 3885 pg/mL in the IL-10-2849AA carriers and 4559 pg/mL in the IL-10-2849AG/GG carriers, indicating that IL-10-2849AA carriers have a reduced IL-10 production of approximately 15%. The results for the rs1800896, rs1800871, and rs6693899 polymorphisms were as follows: *p* = 0.63, *p* = 0.90, and *p* = 0.55) [20]. It was found that rs1800896 in the promoter region of *IL-10* increased the production of IL-10. German researchers decided to investigate this polymorphism. Their study included 181 MS patients and 85 healthy volunteers. However, no association was found between the IL-10 rs1800896 genotypes and the severity of clinical onset or MS (*p* = 0.056) [21].

In our study, we also examined IL-10 levels and compared them between MS patients and healthy individuals. In total, 26 subjects with MS and a control group of 19 subjects were selected for the study. There was no statistically significant difference in the effect of the IL-10 concentration in the MS group (median (IQR): 0.828 (1.533) vs. 0.756 (0.528), *p* = 0.872). No statistically significant differences were found between the IL-10 levels in the healthy subjects and the MS patients with respect to age, sex, and influence on MS development. We found no association between the *IL-10* rs1800871, rs1800872, and rs1800896 genotypes and levels with disease clinic.

The changes in IL-10 values have been studied by other authors. Silva et al. note that the reference values for IL-10 in individuals with MS are scarce [22]. According to Kwilasz et al., the IL-10 levels in peripheral-blood mononuclear cells are lower in individuals with MS compared to healthy controls. It is noteworthy that the secretion of IL-10 by these cells is reduced before relapse and increases during remission, suggesting that IL-10 is necessary for recovery to occur [23]. Molina et al. reported similar data, noting that IL-10 inhibits the expression of pro-inflammatory cytokines, thereby neutralizing the inflammatory process. Therefore, strategies aimed at increasing IL-10 may be effective in the treatment of autoimmune diseases, such as MS [24].

The study by Trenova et al. investigated the association between tumor necrosis factor (TNFα) alpha, interleukin (IL)-17A, IL-18, IL-10 protein levels, and cognitive impairment in patients with relapsing-remitting MS. It was found that TNFα; (*p* < 0.0001) and (IL)-17A; (*p* = 0.038) increased serum concentration and decreased IL-10 protein concentration; (*p* = 0.024) in patients with MS, impairing the patients’ cognitive function [25].

Sedeeq et al. studied IL-10 in relapsing-remitting (RR) MS in 32 RRMS (relapsing-remitting multiple sclerosis) and 26 healthy control subjects and demonstrated that IL-10 in remission was positively correlated with disease duration (r = 0.41; *p* = 0.02) [26].

Animal studies analyzing IL-10 as protective and IL-6 as disease-promoting were also performed in a mouse model of MS and EAE. In this model, IL-10-deficient mice developed a more severe form of EAE compared to wild-type mice, while the mice overexpressing IL-10 were resistant to EAE [27,28]. The ability of recombinant IL-10 or IL-10-transduced cells to control the disease once established yielded conflicting results [29,30,31,32]. These results suggest that not only does the presence of IL-10 limit the disease, but that regulatory cells expressing IL-10 are key to controlling abnormal immune responses.

## 5. Conclusions

The strength of this study was that, for the first time, we analyzed *IL-10* rs1800871, rs1800872, and rs1800896 and IL-10 serum levels in patients with multiple sclerosis in the Baltic population and compared them with healthy controls without other diseases.

In future research, more patients with MS should be included, and IL-10 serum levels should be evaluated, depending on disease stages and treatment modalities, both after and before treatment.

## Figures and Tables

**Figure 1 brainsci-12-00800-f001:**
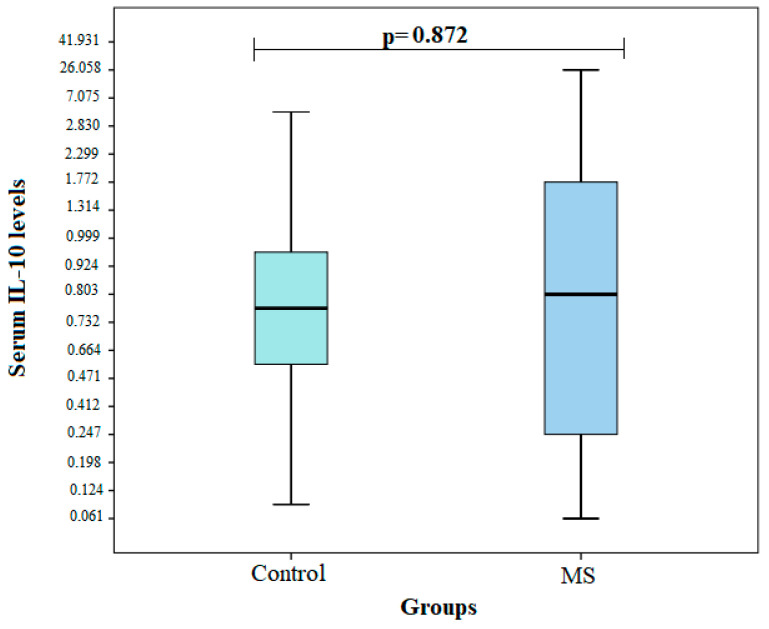
Interleukin-10 concentrations between the groups.

**Table 1 brainsci-12-00800-t001:** Characteristics of study subjects.

Characteristics	Group		*p*-Value
	I Group: Subjects with MS (*n* = 148)	II Group: Control Group (*n* = 213)	
Men, *n* (%)	54 (36.5)	65 (30.5)	0.235
Women, *n* (%)	94 (63.5)	148 (69.5)	
Age, median (IQR)	36 (16)	31 (19)	0.269

**Table 2 brainsci-12-00800-t002:** *IL-10* rs1800871, rs1800872, and rs1800896 genotype and allele frequencies in multiple sclerosis patients and control groups.

SNP	Genotype/Allele	MS Group (*n* = 148) N (%)	Control Group (*n* = 213) N (%)	*p*-Value	HWE *p*-Value
*IL-10* rs1800871	GG	89 (60.1)	118 (55.4)	0.547	0.946
	AG	53 (35.8)	82 (38.5)		
	AA	6 (4.1)	13 (6.1)		
	In total:	148 (100)	213 (100)		
	Allele: G A	231(78) 65 (22)	318 (74.6) 108 (25.4)	0.294	
*IL-10* rs1800872	GG	87 (58.8)	119 (55.9)	0.785	0.869
	TG	54 (36.5)	81 (38.0)		
	TT	7 (4.7)	13 (6.1)		
	In total:	148 (100)	213 (100)		
	Allele: G T	228 (77) 68 (23)	319 (74.9) 107 (25.1)	0.508	
*IL-10* rs1800896	TT	42 (28.4)	75 (35.2)	0.312	0.207
	TC	71 (48.0)	98 (46.0)		
	CC	35 (23.6)	40 (18.8)		
	In total:	148 (100)	213 (100)		
	Alelle: T C	155 (52.4) 141 (47.6)	248 (58.2) 178 (41.8)	0.119	

**Table 3 brainsci-12-00800-t003:** Comparison of *IL-10* rs1800871, rs1800872, and rs1800896 polymorphisms in multiple sclerosis patients by age.

SNP	Genotype/Allele	≤30 Age		*p*-Value	>30 Age		*p*-Value
		MS Group (*n* = 49) N (%)	Control Group (*n* = 105) N (%)		MS Group (*n* = 99) N (%)	Control Group (*n* = 108) N (%)	
*IL-10* rs1800871	GG	25 (51.0)	58 (55.2)	0.532	64 (64.7)	60 (55.6)	0.407
	AG	22 (44.9)	39 (37.1)		31 (31.3)	43 (39.8)	
	AA	2 (4.1)	8 (7.6)		4 (4.0)	5 (4.6)	
	Allele: A G	72 (73.5) 26 (26.5)	155 (73.8) 55 (26.2)	0.949	159 (80.3) 39 (19.7)	163 (75.5) 53 (24.5)	0.237
*IL-10* rs1800872	GG	24 (49.0)	58 (55.2)	0.430	63 (63.6)	61 (56.5)	0.522
	GT	23 (46.9)	39 (37.1)		31 (31.3)	42 (38.9)	
	TT	2 (4.1)	8 (7.6)		5 (5.1)	5 (4.6)	
	Allele: G T	71 (72.4) 27 (27.6)	155 (73.8) 55 (26.2)	0.801	157 (79.3) 41 (20.7)	164 (75.9) 52 (24.1)	0.412
*IL-10* rs1800896	TT	18 (36.7)	43 (41.0)	0.883	24 (24.2)	32 (29.6)	0.580
	TC	23 (46.9)	46 (43.8)		48 (48.5)	52 (48.1)	
	CC	8 (16.3)	16 (15.2)		27 (27.3)	24 (22.2)	
	Allele: C T	59 (60.2) 39 (39.2)	132 (62.9) 78 (37.1)	0.655	96 (48.5) 102 (51.5)	116 (53.7) 100 (46.3)	0.289

**Table 4 brainsci-12-00800-t004:** *IL-10* rs1800871, rs1800872, and rs1800896 genotype and allele frequencies in MS patients and controls by gender.

SNP	Genotype/Allele	Men		*p*-Value	Women		*p*-Value
		MS Group (*n* = 54) %	Control Group (*n* = 65) %		MS Group (*n* = 94) Proc.	Control Group (*n* = 148) %	
*IL-10* rs1800871	GG	33 (61.1)	36 (55.4)	0.751	56 (59.6)	82 (55.4)	0.463
	AG	18 (33.3)	26 (40.0)		35 (37.2)	56 (37.8)	
	AA	3 (5.6)	3 (4.6)		3 (3.2)	10 (6.8)	
	Allele: A G	84 (77.8) 24 (22.2)	98 (75.4) 32 (24.6)	0.665	147 (78.2) 41 (21.8)	220 (74.3) 76 (25.7)	0.333
*IL-10* rs1800872	GG	30 (55.6)	36 (55.4)	0.795	57 (60.6)	83 (56.1)	0.453
	TG	20 (37.0)	26 (40.0)		34 (36.2)	55 (37.2)	
	TT	4 (7.4)	3 (4.6)		3 (3.2)	10 (6.8)	
	Allele: G T	80 (74.1) 28 (25.9)	98 (75.4) 32 (24.6)	0.817	148 (78.7) 40 (21.3)	221 (74.7) 75 (25.3)	0.306
*IL-10* rs1800896	TT	16 (29.6)	19 (29.2)	0.987	26 (27.7)	56 (37.8)	0.204
	TC	25 (46.3)	31 (47.7)		46 (48.9)	67 (45.3)	
	CC	13 (24.1)	15 (23.1)		22 (23.4)	25 (16.9)	
	Allele: C T	57 (52.8) 51 (47.2)	69 (53.1) 61 (46.9)	0.963	98 (52.1) 90 (47.9)	179 (60.5) 117 (39.5)	0.070

**Table 5 brainsci-12-00800-t005:** Genotype distribution and IL-10 concentration.

Genotype	IL-10 Serum Level		*p* Value *
	MS Median (IQR)	Control Median (IQR)	
rs1800871			
GG	0.698 (1.445)	0.817 (1.844)	0.699
GA + AA	0.828 (1.703)	0.756 (0.304)	0.934
rs1800872			
GG	0.924 (1.366)	0.817 (1.844)	0.828
GT + TT	0.732 (2.245)	0.756 (0.304)	0.908
rs1800896			
TT	0.732 (1.410)	0.756 (0.304)	0.895
TC+CC	0.924 (1.525)	0.817 (4.823)	0.628

* Mann–Whitney U test used.

**Table 6 brainsci-12-00800-t006:** Linkage disequilibrium between every two *IL-10* SNPs.

SNPs	MS vs. Controls		
	D’	r^2^	*p*-Value
rs1800871—rs1800872	0.9459	0.8813	0.0
rs1800871—rs1800896	0.9434	0.2222	0.0
rs1800872—rs1800896	0.9064	0.2080	0.0

**Table 7 brainsci-12-00800-t007:** Associations between *IL-10* haplotypes and risk of multiple sclerosis.

Haplotype	rs1800871	rs1800872	rs1800896	Frequency (%)			OR (95% CI)	*p*-Value
				MS	Controls	Total		
1	G	G	C	45.22	41.55	42.99	1.00	---
2	G	G	T	30.06	32.86	31.79	0.85 (0.60–1.20)	0.36
3	A	T	T	19.25	24.88	22.58	0.75 (0.50–1.11)	0.15
rare	*	*	*	NA	NA	2.64	5.99 (1.68–21.34)	0.006

OR—odds ratio; CI—confidence interval; p—significance level when *p* = 0.05; AIC—Akaike information criteria; NA—not applicable; * rare—pooled haplotypes with frequencies <1%.

## Data Availability

The data used to support the findings of this study are available from the corresponding author upon request.

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
