# Peer review of "IL-10* Gene Polymorphisms and IL-10 Serum Levels in Patients with Multiple Sclerosis in Lithuania"

_brainsci, 2022, doi:10.3390/brainsci12060800_

Round 1

Reviewer 1 Report

Review for the article “IL-10 rs1800871, rs1800872, and rs1800896 and IL-10 serum levels in patients with multiple sclerosis in Lithuania

The article entitled “IL-10 rs1800871, rs1800872, and rs1800896 and IL-10 serum levels in patients with multiple sclerosis in Lithuania” studied the relationship between three genetic variation rs1800871, rs1800872, and rs1800896 located in the IL-10 gene and the risk for multiple sclerosis (MS). Also, the authors determined the IL-10 levels in patients and controls serum, trying to establish a link with MS.

The results obtained by the authors confirmed that IL-10 rs1800896 polymorphism could be a genetic risk factor for MS in women.

Comments

  1. In the introduction section the authors mentioned the three polymorphism in the IL -10 gene: rs1800871, rs1800872, and rs1800896. Please described them (nucleotide position, and the nucleotide bases involved).

2.In paragraph 67, the authors said “Diagnosis of MS confirmed with 2017 diagnostic criteria [12]”. Please described the parameters used in order to establish de diagnosis of MS.

  1. The authors mentioned the exclusion criteria for patients. Did the patients have other autoimmune diseases?
  2. Please give the reference for the salting out method used for DNA isolation. Please give details for the real-time PCR methods, in order to be reproduce in other laboratories (e.g. sequences of the primers).
  3. Please give details regarding demographic, lifestyle (e.g. smoking, alcohol consumption, etc) and clinical (the form of multiple sclerosis, age in the moment of diagnosis) characteristics for the patients and controls. Have the patients neurological examination? Please detailed.
  4. Did the patients receive any treatment, or they are new patients?
  5. Line 264: “No statistically significant differences were found between IL -10 levels in healthy subjects and MS patients with respect to age, sex, and influence on clinical presentation of MS. We found no association between IL -10 rs1800871, rs1800872, and rs1800896 266 genotypes and levels with disease clinic.”

What does it mean “disease clinic”?

I saw that the genetic analysis was performed in 127 patients with MS and 195 controls. Why the authors determine the IL-10 levels only in 19 patients with MS and 29 controls. I think that the results are not relevant. How did you select these subjects?

  1. The authors used bivariate analysis. Maybe it is interesting to use also multivariate analysis if you obtained statistically significance using univariate analysis.
  2. Maybe it is interesting to check for the association between IL-10 polymorphisms and clinical characteristics of the patients, in both women and men.
  3. Line 277: The authors present the results obtained by Trenova et al regarding the association between IL-10 levels and relapsing remitted multiple sclerosis (RRMS), but they didn’t do this in their article. Maybe it will be interesting to analyse this hypothesis and also the association with other forms of MS.

Also, the authors present in this line the results regarding other genes in MS, that are not the subjects of this study.

  1. Please indicate the equipment that you used for IL-10 determination and also the sensitivity of the method.

Author Response

Dear Editor and Reviewers,

We kindly appreciate the revision of our manuscript. We have highlighted the changes we made in the manuscript by using the track changes mode in MS Word. Hope that the revised manuscript will be acceptable for publication in your journal. Enclosed please also find attached our point-by-point response to the comments raised by the reviewers (editors).

Reviewer 2 Report

Introduction

As required by the guidelines for authors, the introduction briefly describes how the authors constructed the hypothesis of their work, but this detracts from the clarity of the sequence of discourse, and could increase the literature references supporting the choice of the three polymorphisms, beyond the citations in the materials and mathods section "2.2 Polymorphism Selection ".

In contrast, the discussion is very well described and the results well argued.

-Xing Li et al Association of Polymorphisms in Inflammatory Cytokines Encoding Genes With Anti-N-methyl-D-Aspartate Receptor Encephalitis in the Southern Han Chinese. Front. Neurol., 11 December 2020 | https://doi.org/10.3389/fneur.2020.553355

2. Materials and Methods

2.1. Study Subjects

-Line 67 - replace the line with “MS diagnosis confirmed with McDonald diagnostic criteria updated in 2017”

We found that the IL -10 rs1800896 T allele may play a predominant role in the development of MS in women. No statistically significant differences were found in the male group.

2.4. Quantification of IL-10 Serum Levels

…..briefly describe the method……

3. Results

In the dermographic table (Table 1. Characteristics of study subjects) add EDSS and disease duration, because in the discussion the authors state, "........ that the IL -10 rs1800896 T allele may play a predominant role in the development of MS in women. No statistically significant differences were found in the male group. "...... It would be clearer to better describe the characteristics of the groups.

3.1. IL-10 Serum Levels in MS and Control Groups

.....non says how many sera from male patients and how many from female patients....

The sera analyzed do not seem to have a numerosity necessary for comparison.

Author Response

(The authors gave the same response as above.)

Reviewer 3 Report

Masilionyte U et al. IL-10 rs1800871, rs1800872, and rs1800896 and IL-10 serum levels in patients with multiple sclerosis in Lithuania

The article is of interest to the scientific community. Although there are few issues that need to be addressed.

1. IL-10 used in the title is too redundant. Could the author rephrase the title? Similarly, there are phrases with redundancies, which need to be changed. 

2. SNPs selection criterion of IL-10 is not clear.

3. There is a trend of significance for the age group between healthy control and MS patients. It is important to do age-correction to rest other data just to nullify the effect of age in the differences of IL-10 levels. Although it is not significant.

4. There is a need to improve writing style so as to make it easy to follow.

5. Is there a possibility of increasing the sample size? If not, please calculate the power of the study.

Author Response

(The authors gave the same response as above.)

Round 2

Reviewer 3 Report

Comments has been answered correctly.